# Sequential Intraoperative Evaluation of Rotational Stability of Cementless Stem in Total Hip Arthroplasty: A Broach-Based Study

**DOI:** 10.3390/jcm12175444

**Published:** 2023-08-22

**Authors:** Sakae Kinoshita, Koichi Kinoshita, Tetsuya Sakamoto, Hajime Seo, Masahiro Suzuki, Takuaki Yamamoto

**Affiliations:** Department of Orthopaedic Surgery, Faculty of Medicine, Fukuoka University, 7-45-1 Nanakuma, Jonan-ku, Fukuoka 814-0180, Japan; sakae.k220@gmail.com (S.K.); kinopfukuoka@yahoo.co.jp (K.K.); tetsusakamotoplayer@yahoo.co.jp (T.S.); hajime@gmail.com (H.S.); 0321suzuki@gmail.com (M.S.)

**Keywords:** rotational stability, cementless stem, torque wrench, broach

## Abstract

In cementless total hip arthroplasty, the rotational stability of the stem is generally confirmed in a one-time manual evaluation by the surgeon. This study was performed to evaluate the sequential intraoperative changes in rotational stability using a torque wrench. Primary total hip arthroplasty was performed on 52 consecutive hips using a single wedge stem design. Intraoperative evaluation of rotational stability was uniformly performed using a torque wrench. Evaluations were performed immediately after broach insertion and immediately before the final insertion of the stem (after placement of the acetabular cup). Immediately after the insertion of the broach, rotational stability was checked and confirmed to be fully stabilized using a torque wrench in all cases, and the stability was maintained in 17 of 52 (33%) hips immediately before the final insertion of the stem. Among the hips showing instability, 11 of 35 (31%) broaches were upsized, while the remaining 24 hips achieved stabilization through deeper insertion of the broach. In conclusion, the rotational stability achieved immediately after the insertion of the broach was not necessarily maintained during surgery, indicating that rotational stability may need to be checked at multiple time points intraoperatively.

## 1. Introduction

Total hip arthroplasty (THA) is one of the most commonly performed orthopedic surgical procedures and has a high success rate [1]. Cementless femoral stems with many different designs have been used, and the reported survival rates are excellent [2,3,4,5]. However, stem subsidence reportedly occurs in 1.6% to 10.9% of cases 6 to 8 weeks after surgery [6,7]. In addition, several gene pathways and genes have been proposed to contribute to individual susceptibility to aseptic loosening following THA [8].

The initial stability of a cementless stem is achieved through proximal or distal fixation, which depends on press–fit fixation to the cortical bone in each stem design [9,10]. The cavity is prepared prior to the implantation of the stem by extracting or compacting the trabecular bone using a broach. The compaction of trabecular bone enhances primary stem stability by reducing micromotion at the bone–stem interface and increasing the fixation strength of the stem in the initial period [11,12].

Regarding initial stem stability, previous studies have shown that low systemic bone mineral density (BMD), geometric changes in the proximal femur, and aging may increase the initial migration of cementless femoral stems [13].

Although various reports have addressed postoperative stem subsidence and its causes, no reports have focused on the relationship between the intraoperative evaluation of rotational stability and postoperative stem subsidence. The rotational stability of the broach is generally confirmed through a one-time manual evaluation by the surgeon. However, even when rotational stability is present at the time of the intraoperative evaluation, its sequential changes during surgery have not been evaluated. This study was therefore performed to evaluate sequential intraoperative rotational stability using a torque wrench and examine the possible causes of intraoperative and postoperative instability.

## 2. Materials and Methods

### 2.1. Patients

This study was approved by our institutional review board at Fukuoka University (approval number: U20-07-009). All methods were performed in accordance with the relevant guidelines and regulations. Both written and verbal informed consent were obtained from all participants in this study. Prior to commencement of the study, the single hip surgeon involved in the study had performed primary THA through the anterolateral approach in the supine position in more than 150 hips.

We prospectively collected data for 55 consecutive hips of 54 patients who underwent primary THA using the anterolateral approach by the same surgeon from September 2020 to December 2021. The single-wedge stem design (Initia; Kyocera Corporation, Kyoto, Japan) defined in the classification established by Khanuja et al. [9] was used in all cases. One month prior to the intervention, BMD was measured in all patients using standard procedures through dual-energy X-ray absorptiometry (Horizon A; Hologic Inc., Marlborough, MA, USA). Participants who met either of the two following criteria were prohibited from undergoing a DXA scan [14]: pregnancy (positive urine pregnancy test and/or self-reported positive test at the time of DXA examination [15]), or performance of a DXA scan at another hospital within the past 4 months. Patients who lacked BMD measurement in the proximal femur on the operative side were excluded from the study. Three hips of three patients were excluded. Therefore, 52 hips of 51 patients were included. Forty-six hips were diagnosed with osteoarthritis, and six hips were diagnosed with osteonecrosis. The broach and stem design of the implant are shown in Figure 1. The characteristics of the patients are shown in Table 1.

### 2.2. Surgical Planning and Technique

During the preoperative preparations, preoperative planning software (mediCAD Hectec GmbH, Altdorf, Germany) was used to select the best stem size to fit and fill the femur morphology of each patient.

The surgical procedures were performed via the anterolateral approach with the patient in the supine position without the use of a traction table [16]. The implant was placed using the stem-first technique, and the selected broach was inserted into the femoral bone cavity in alignment with the anatomical structure. Femoral broaching was performed by manual insertion with a mallet to engage the medial calcar and lateral cortex of the femur.

Immediately after the broach size was confirmed to be adequate through fluoroscopy or intraoperative X-ray, rotational stability was evaluated using a torque wrench (Tohnichi Mfg. Co., Ltd., Tokyo, Japan), which was used to constantly apply a rotational moment of 7 Nm (Figure 2). This rotational moment was selected because previous studies have shown that a rotational moment of 7.2 Nm is the maximum threshold for the blades of the broach to penetrate the trabecular bone [17]. The acetabular cup was then placed after the broach had been inserted. Finally, the stem was inserted after the rotational stability and leg length were confirmed.

All patients commenced active motion exercises and full weight-bearing training on the second postoperative day.

### 2.3. Evaluation of Rotational Stability

Sufficient rotational stability was defined as movement of the torque wrench by <1 mm from the central point of the medial side of the broach in the direction of rotation. Rotational stability was evaluated at two time points: immediately after the insertion of the broach and immediately before the final insertion of the stem (after placement of the acetabular cup). If rotational instability was observed immediately after broach insertion, the broach was inserted more deeply with further impaction or a larger broach was selected until rotational stability was obtained using the torque wrench. Rotational stability was then checked immediately before the final insertion of the stem and classified into stable and unstable groups. If rotational instability of the broach was observed immediately before the final insertion of the stem, the broach was inserted more deeply with further impaction or a larger broach was selected until rotational stability was obtained. Once rotational stability was achieved, the stem was inserted.

### 2.4. Radiographic Evaluation

The cortical index was calculated from anteroposterior plain radiographs of the hip using the method described by Dorr et al. [18]. The cortical index was obtained by dividing the thickness of the femoral bone cortex at a point 10 cm distal to the center of the lesser trochanter (i.e., the length of the lateral aspect of the femoral bone cortex minus the length of the medullary cavity) by the lateral extent of the femoral bone cortex. The morphology of the proximal femur was classified as Dorr type A, B, or C [18], which corresponded to measured cortical indices of ≥0.58, 0.49–0.57, and ≤0.48, respectively, according to the description by Nakaya et al. [19]. Stem subsidence was evaluated 1, 4, and 12 weeks after surgery. Subsidence of at least 2 mm on postoperative radiographs was considered clinically significant and recorded as positive subsidence [20]. Subsidence of the femoral stem was measured as the distance between the proximal aspect of the greater trochanter and shoulder of the femoral stem [21]. The method used to measure stem subsidence is shown in Figure 3. Femoral BMD scanning was performed using a Hologic Discovery A densitometer with Apex software (Apex Version 5.6.0.4. Hologic Inc., Marlborough, MA, USA). BMD was selected in four regions of the femur: the femoral neck, trochanter, intertrochanteric region, and total femur [22].

### 2.5. Statistical Analysis

Statistical analyses were performed using SPSS software version 20.0 (IBM Corp., Armonk, NY, USA). The chi-square test was conducted to assess the effect of the Dorr type and preoperative diagnoses of osteoarthritis and osteonecrosis on rotational stability immediately before final insertion of the stem. The Mann–Whitney U test was conducted to evaluate the effects of BMD in the four selected regions of the femur, intraoperative bleeding, the cortical index, and the interval between the first and second evaluation time points (immediately after broach insertion and immediately before final insertion of the stem) on rotational stability. A *p* value of <0.05 was considered statistically significant.

## 3. Results

Immediately after the insertion of the broach, rotational stability was confirmed using a torque wrench in all cases. Immediately before the final insertion of the stem, rotational stability was maintained in 17 of 52 (33%) hips, while 35 hips showed instability. Among the hips that showed instability immediately before the final insertion of the stem, 11 of 35 (31%) broaches were upsized, while the remaining hips were rotationally stabilized through deeper insertion. The BMD in the four regions of the femur, intraoperative bleeding, the Dorr type, and the interval between the first and second evaluation time points were not significantly different between the two groups (Table 2). There was no significant difference in the preoperative diagnosis of osteonecrosis and osteoarthritis between the two groups (*p* = 0.97).

Stem subsidence of ≥2 mm occurred in two hips (2.5 and 2.9 mm) 1 week after surgery and in three hips (2.5, 3.1, and 3.2 mm) 4 weeks after surgery. There was no further subsidence at 12 weeks postoperatively. The BMD in the four regions of the femur was compared between hips with and without postoperative stem subsidence, but there were no significant differences (Table 3).

## 4. Discussion

The compaction of trabecular bone by broaching enhances the primary stem stability by reducing micromotion at the bone–stem interface and increasing the fixation strength of the stem in the initial period [11,12]. This fixation is thought to result from the spring-back effect caused by the viscoelastic behavior of the compacted trabecular bone [23,24]. However, excessive loading may cause inelastic deformation of the bone, which can lead to microdamage, fracture, compaction, and crushing [25]. A rotational moment of at least 7.2 Nm with the broach inserted into the femur has been reported to destroy trabecular bone and destabilize the broach [17]. In this study, rotational stability was not maintained throughout the surgery in some cases. In addition to the viscoelastic behavior and stress relaxation of bone, the rotational moment and force applied to the broach, such as those in hip reduction and dislocation as well as hip stability testing, may also cause such rotational instability. The results may suggest the necessity of intraoperative multipoint verification of stability.

Regarding fixation by the broach design, the rotational stability of the single-wedge design is achieved mainly through the proximal contact of the broad, flat shape and wedge fixation in the medial–lateral plane [26,27]. The implant is primarily in contact with compressed trabecular bone, which may result in reduced rotational stability if the rotational moment destroys trabecular bone. In our study, the rotational stability was maintained in only 30% of hips. Meanwhile, BMD, preoperative diagnosis, intraoperative bleeding, and time interval did not significantly differ between the broaches that were stable and unstable. These results suggest that the rotational moment of the force acting on the broach may affect rotational stability and that intraoperative procedures may affect broach instability. Trabecular bone may have been destroyed by the strong rotational force applied to the broach during reduction and dislocation of the hip and hip stability testing.

In this study, the Dorr types A, B, and C and cortical index of the femur did not significantly differ between the broaches that were stable and unstable. With respect to femoral morphology and stability, a previous study showed that over a 10-year follow-up, the single-wedge cementless stem was a recommendable option with satisfactory outcomes and excellent stem survivorship regardless of Dorr types A, B, and C [28]. Additionally, no stem subsidence of >3 mm was observed. Therefore, in single-wedge stems, initial fixation and rotational stability may be achieved by selecting the appropriate sizing for wedge fixation in the proximal medial–external plane with the proximal medullary cavity, regardless of the medullary cavity morphology. In this study, we were unable to demonstrate the factors that caused rotational instability before the final insertion of the stem; however, inquiring into other causes and increasing the number of cases may reveal these factors in future studies.

Previous studies have shown that patients with low systemic BMD have higher subsidence of the femoral stem during the first 3 months after surgery than those with normal BMD [13]. In this study, the incidence of stem subsidence up to 12 weeks postoperatively in single-wedge stems (9.6%) was similar to that in previous reports. In addition, BMD did not affect the intraoperative rotational stability of the broach or postoperative stem subsidence within 3 months, despite the fact that most of the patients were postmenopausal osteoporotic women with low BMD. This suggests that factors other than low BMD may affect the subsidence. Previous studies have shown that excessive drilling and grinding to ensure the fitting of a component mechanically and thermally damage bone and inhibit bone growth [29]. Stability evaluation with the proper rotational force using a torque wrench may prevent excessive drilling and grinding.

This study had several limitations. First, rotational forces applied to the broach during surgery could not be evaluated in detail. Second, because rotational stability was not evaluated after stem insertion, it was not possible to assess whether rotational stability had been completely achieved even after the final stem insertion. Third, the postoperative stem subsidence was not compared between cases in which a torque wrench was and was not used. Fourth, the cause of rotational instability could not be identified. In addition to the viscoelastic behavior and stress relaxation of bone, other factors associated with rotational instability during surgery may need to be further evaluated.

## 5. Conclusions

The rotational stability achieved immediately after the insertion of the broach was not necessarily maintained during surgery, indicating the necessity of multipoint intraoperative checks of the rotational stability.

## Figures and Tables

**Figure 1 jcm-12-05444-f001:**
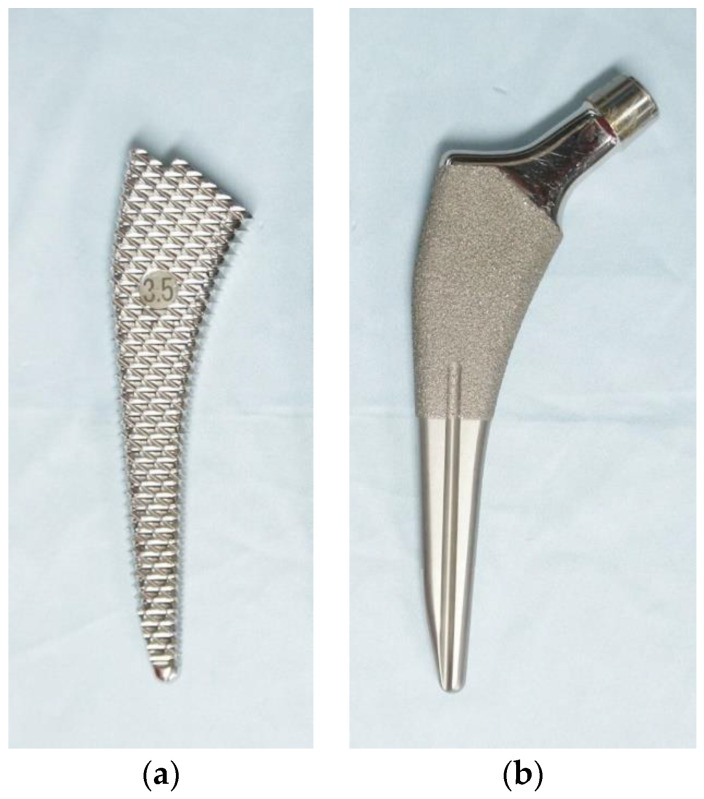
Anterior views of (**a**) broach and (**b**) stem with single-wedge design. The stem is 0.25 mm smaller on both the medial and lateral sides than the broach, but the coating on the distal 10 mm length of the sprayed area increases in a gradient from distal (0.2 mm) to proximal (0.5 mm). The stem and broach were manufactured by Kyocera Corporation (Kyoto, Japan).

**Figure 2 jcm-12-05444-f002:**
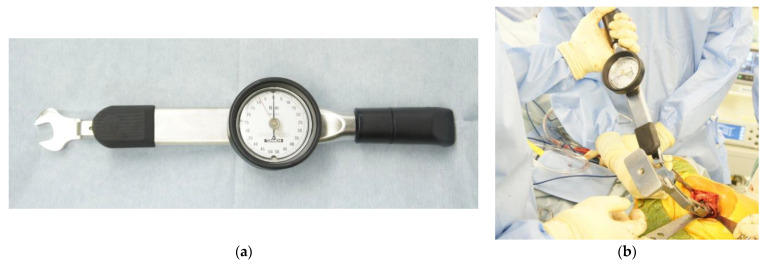
Torque wrench and evaluation of rotational stability. (**a**) Torque wrench (Tohnichi Mfg. Co., Ltd., Tokyo, Japan) used to check rotational stability. (**b**) Photograph of the intraoperative evaluation of rotational stability after broach insertion.

**Figure 3 jcm-12-05444-f003:**
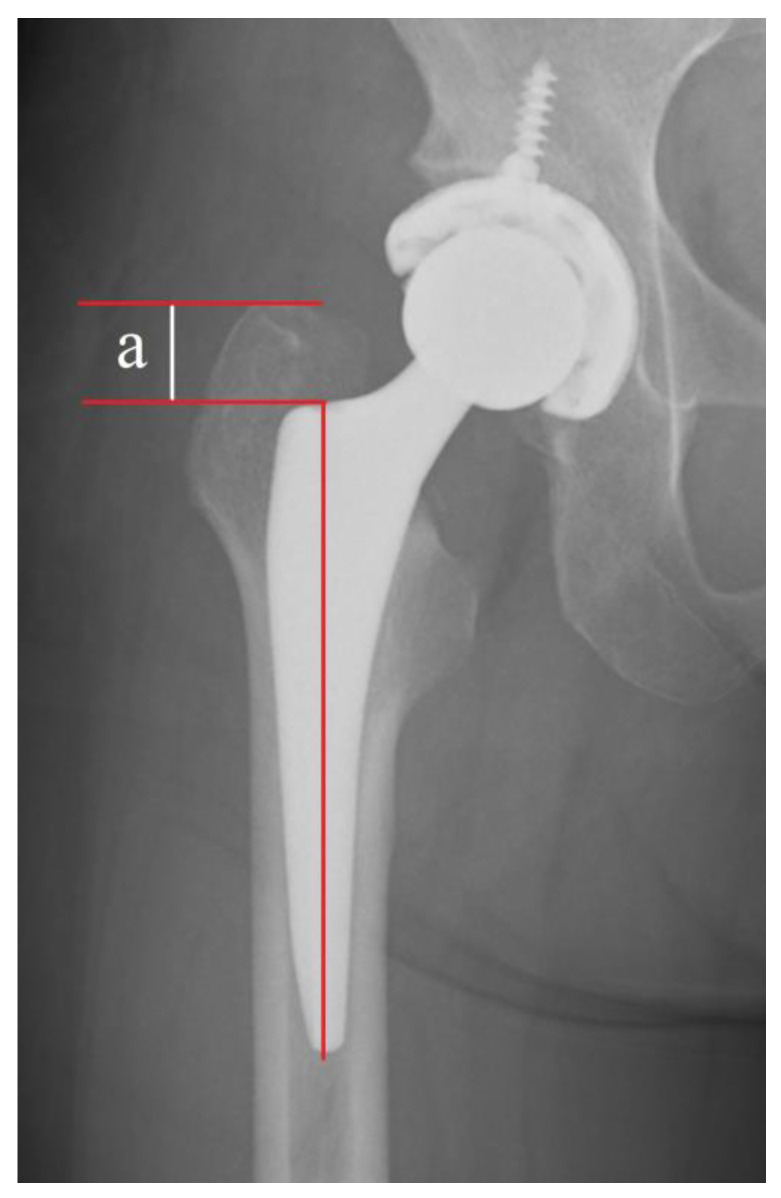
Radiographic measurement of femoral stem subsidence. The white line (a): Subsidence was measured as the distance between the proximal aspect of the greater trochanter and shoulder of the femoral stem on calibrated anteroposterior X-rays of the hip. The vertical red line: coronal stem alignment.

**Table 1 jcm-12-05444-t001:** Characteristics of 52 hips of 51 patients treated through primary total hip arthroplasty with anterolateral approach in supine position.

	Single Wedge (*n* = 52 Hips)
Sex, male:female (no. of hips)	9:43
Age (yr)	63.3 ± 11.7 (39–87)
Height (cm)	155.0 ± 7.2 (141.5–177.0)
Weight (kg)	61.7 ± 12.0 (40.8–92.0)
BMI (kg/m^2^)	25.6 ± 3.9 (19.0–35.6)

Values are shown as mean ± standard deviation (range) unless otherwise indicated. BMI, body mass index.

**Table 2 jcm-12-05444-t002:** BMD, intraoperative bleeding, cortical index, time interval, and Dorr type for broach in each group.

	Unstable Group (*n* = 35 Hips)	Stable Group (*n* = 17 Hips)	*p* Value
BMD of total femur (g/cm^2^)	0.79 ± 0.14 (0.56–1.21)	0.74 ± 0.14 (0.54–1.14)	0.25
BMD of femoral neck (g/cm^2^)	0.73 ± 0.12 (0.62–1.21)	0.73 ± 0.15 (0.52–1.10)	0.89
BMD of trochanter (g/cm^2^)	0.60 ± 0.13 (0.41–0.94)	0.55 ± 0.12 (0.37–0.90)	0.23
BMD of intertrochanteric region (g/cm^2^)	0.80 ± 0.18 (0.62–1.18)	0.80 ± 0.20 (0.6–1.10)	0.99
Intraoperative bleeding (mL)	310 ± 218 (44–611)	404 ± 164 (43–739)	0.10
Cortical index	0.51 ± 0.09 (0.37–0.61)	0.49 ± 0.06 (0.33–0.67)	0.26
Time interval (min)	38.1 ± 11.6 (26 to 71)	37.4 ± 13.5 (22 to 62)	0.49
Dorr A:B:C (no. of hips)	12:15:8	8:8:1	0.30

Values are shown as mean ± standard deviation (range) unless otherwise indicated. BMD, bone mineral density.

**Table 3 jcm-12-05444-t003:** BMD for implants that had subsided by 12 weeks postoperatively and those that had not.

	No Subsidence Group (*n* = 47 Hips)	Subsidence Group (*n* = 5 Hips)	*p* Value
BMD of total femur (g/cm^2^)	0.77 ± 0.15 (0.54–1.21)	0.78 ± 0.10 (0.62–0.85)	0.37
BMD of femoral neck (g/cm^2^)	0.74 ± 0.13 (0.52–1.21)	0.67 ± 0.08 (0.57–0.77)	0.17
BMD of trochanter (g/cm^2^)	0.59 ± 0.14 (0.37–0.94)	0.58 ± 0.09 (0.48–0.71)	0.33
BMD of intertrochanteric region (g/cm^2^)	0.88 ± 0.17 (0.60–1.18)	0.91 ± 0.13 (0.72–1.05)	0.47

Values are shown as mean ± standard deviation (range). BMD, bone mineral density.

## Data Availability

The datasets used and/or analyzed during the current study are available from the corresponding author on reasonable request.

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
