# Peer review of "Sequential Intraoperative Evaluation of Rotational Stability of Cementless Stem in Total Hip Arthroplasty: A Broach-Based Study"

_jcm, 2023, doi:10.3390/jcm12175444_

Round 1
Reviewer 1 Report
The study design of this article is basically reasonable, but the clinical reference value is limited, with some innovation, and it is recommended to continue to add relevant research content.
1. “Verbal informed consent was obtained from all participants in this study. ”--How to ensure that patients understand the full extent of the experiment?Whether written informed consent is more reasonable
2. “One month prior to the intervention, BMD was measured in all patients using standard procedures by dual-energy X-ray absorptiometry (Horizon A; Hologic Inc., Marlborough, MA, USA).” --Is there a requirement for BMD in the inclusion criteria?
3. “BMD was measured in four regions of the femur: the femoral neck, trochanter, intertrochanteric region, and total femur. 133” --How can dual-energy X-ray measure these four areas?
4. “Subsidence of at least 2 mm on postoperative radiographs was considered clinically significant and recorded as positive subsidence”--Suggested additional imaging illustrations.
5. The title of the article is "Sequential Intraoperative Evaluation of Rotational Stability of Cementless Stem in Total Hip Arthroplasty" but not evaluated after stem insertion, is incomplete.
6. The data in Table 2 do not match the narrative of the results
7. The prosthesis in this study was fixed proximally, so why did the different groupings of medullary cavity morphology(Dorr A/ B/C) have no significant effect on stability?
8. There were no clear inclusion and exclusion criteria for this study, and whether different diseases have an effect on the stability of the femoral stem?
Need native English polishing.
Reviewer 2 Report
Dear Editor and Authors.
I find this paper as an interesting. However, the paper: line 57:
"Verbal informed consent was obtained from all partici- 57 pants in this study."
All the patients have to sign the agreement before the surgery. Where the information about the new technique of shaft stability estimation? The oral consent is definetly to small one.
Please check the paper and patient's agreements. Potentially, the double checking may provide to femur breaking.
Round 2
Reviewer 2 Report
After the revision the paper can be publish in the JCM (ISSN 2077-0383)
Kindly regards